# Circulating Tumor DNA as a Complementary Prognostic Biomarker during CAR-T Therapy in B-Cell Non-Hodgkin Lymphomas

**DOI:** 10.3390/cancers16101881

**Published:** 2024-05-15

**Authors:** Sarah Monick, Allison Rosenthal

**Affiliations:** Department of Medicine, Mayo Clinic Arizona, Phoenix, AZ 85054, USA; rosenthal.allison@mayo.edu

**Keywords:** B-NHL, circulating tumor DNA, minimal residual disease, liquid biopsy, CAR-T therapy

## Abstract

**Simple Summary:**

Circulating tumor DNA (ctDNA) is an emerging, multifaceted biomarker for predicting outcomes in B-cell non-Hodgkin lymphomas (B-NHL) following CD19-directed CAR-T therapy. Conventional imaging techniques face challenges in accurately assessing treatment response and detecting early relapse, highlighting the need for non-invasive analytes like ctDNA. By overcoming these limitations, ctDNA offers real-time insights into treatment response, resistance mechanisms, and early relapse detection. Integration of ctDNA monitoring into clinical practice has the potential to personalize therapeutic strategies, optimize patient outcomes, and guide the development of novel therapeutics. However, standardization of assay methods and consensus on clinical response metrics are essential for realizing its full potential in relapsed/refractory (R/R) B-NHL management.

**Abstract:**

The emergence of CD19-directed chimeric antigen receptor T-cell (CAR-T) therapy has revolutionized the treatment paradigm for R/R B-cell NHLs. However, challenges persist in accurately evaluating treatment response and detecting early relapse, necessitating the exploration of novel biomarkers. Circulating tumor DNA (ctDNA) via liquid biopsy is a non-invasive tool for monitoring therapy efficacy and predicting treatment outcomes in B-NHL following CAR-T therapy. By overcoming the limitations of conventional imaging modalities, ctDNA assessments offer valuable insights into response dynamics, molecular mechanisms of resistance, and early detection of molecular relapse. Integration of ctDNA monitoring into clinical practice holds promise for personalized therapeutic strategies, guiding the development of novel targeted therapies, and enhancing patient outcomes. However, standardization of assay methodologies and consensus on clinical response metrics are imperative to unlock the full potential of ctDNA in the management of B-NHL. Prospective validation of ctDNA in clinical trials is necessary to establish its role as a complementary decision aid.

## 1. Introduction

B-cell non-Hodgkin lymphomas (B-NHL) represent a clinically, histologically, and genetically diverse group of hematologic malignancies arising from the mature B-lymphocyte. The approval of CD19-directed chimeric antigen receptor T-cell (CAR-T) therapy has revolutionized the treatment landscape in the relapsed/refractory (R/R) setting, with multi-center trials demonstrating complete response (CR) rates of 40–54%, 67%, and 69–74% in patients with aggressive B-cell lymphomas [1,2,3], mantle cell lymphoma [4], and indolent B-cell lymphomas [5,6], respectively. Currently, four anti-CD19 CAR-T therapies have been approved by the Food and Drug Administration (FDA) for the management of R/R large B-cell lymphoma (LBCL), follicular lymphoma (FL), and mantle cell lymphoma (MCL) including: Axicabtagene Ciloleucel (Yescarta) [7], Brexucabtagene Autoleucel (Tecartus) [8], Lisocabtagene Marleucel (Breyanzi) [9], and Tisagenlecleucel (Kymriah) [10].

The clinical success of CAR-T therapy in inducing high rates of remission poses new challenges in assessing durability of response as nearly 60% of patients treated with CD19 CAR-T therapy will relapse within the first year despite achieving a CR [11]. This is in part due to the persistence of minimal residual disease (MRD), defined as the presence of residual cancer cells after treatment in patients with clinically undetectable disease on standard functional imaging techniques or clinical assessments [12]. Unfortunately, only half of patients with post-CAR-T therapy progression will go on to receive further disease-directed therapies, underscoring the rapid clinical decline of this population [13]. Thus, there is an increasing interest in the use of MRD testing via liquid biopsy as a surrogate marker to better predict response to therapy and detect early relapse, allowing for prompt risk-adapted therapeutic interventions before overt clinical relapse with the aim of improving clinical outcomes. 

In this review, we present an overview of emerging techniques of MRD detection and monitoring in B-cell NHL. Furthermore, we critically examine the utilization of MRD assessments in recent studies following CAR-T therapy and evaluate the evolving role of circulating tumor DNA (ctDNA) as a novel prognostic marker for predicting and detecting early disease relapse.

### Conventional Functional Imaging

Fluorodeoxyglucose positron tomography/computed tomography (FDG PET/CT) imaging using the Deauville 5-point score is the current gold standard in evaluating end-of-treatment response for lymphomas in clinical practice, including after CAR-T therapy [14]. Deauville scoring assesses the maximum standard uptake value (SUV max) of FDG as a surrogate of glucose metabolism compared to liver or mediastinal blood pool reference ranges. An emerging parameter to assess disease activity includes metabolically active tumor volume (MTV), defined as the total three-dimensional volume of the tumor showing increased FDG uptake. MTV differs from SUV in that it represents the spatial extent of FDG uptake by the metabolically active tumor rather than merely quantifying the intensity of metabolic activity. Baseline MTV has shown to be an independent prognostic indicator in B-NHL, with high MTV associated with poor outcomes following chemoimmunotherapy [15,16]. A similar prognostic value in predicting efficacy outcomes after CAR-T therapy has also been demonstrated, with high baseline MTV associated with inferior progression-free survival (PFS) and overall survival (OS) in large B-cell lymphoma (LBCL) patients [17]. More recently, Rojek et al. demonstrated high residual MTV ≥ 106 mL at day 30 post-CAR-T infusion was associated with inferior 1-year PFS (0% vs. 66% for low residual, *p* < 0.01) and 1-year OS (13% vs. 83%, *p* < 0.01) [18]. 

Current American Society for Transplant and Cellular Therapies (ASTCT) guidelines recommend obtaining PET/CT no earlier than 30 days after CAR-T therapy. However, consensus guidelines regarding optimal time points for imaging post-CAR-T therapy have yet to be determined with variation from 1 to 3 months post-CAR-T infusion across clinical trials [3,19,20,21,22]. Interpretation of FDG avid lesions in the trimester following CAR-T therapy is particularly challenging due to suboptimal specificity and sensitivity. Patients can develop pseudo-progression, a phenomenon attributed to ongoing inflammatory responses from immune-mediated killing of cancer cells, leading to high false-positive rates due to an apparent increase in tumor size and activity on PET/CT [23,24,25]. Furthermore, lingering effects of intensive lymphodepleting chemotherapy and high-dose corticosteroids used in the management of toxicities associated with CAR-T therapy including cytokine release syndrome (CRS) or immune effector cell-associated neurotoxicity (ICANs) can lead to the transient response of tumor cells leading to negative results on day 30 PET/CT.

Though achievement of a complete response is essential for cure, approximately 15% of patients with DLBCL with a negative end-of-treatment PET/CT after frontline chemoimmunotherapy will relapse, underscoring the inability to detect MRD and the need for complementary ctDNA assessments to better quantify depth and duration of response [26]. Furthermore, routine image-based surveillance of patients who achieve remission is associated with increased radiation exposure and no demonstrated survival benefit, and is no longer routinely recommended [27,28,29]. Liquid biopsy surveillance assessments with ctDNA have emerged and shown promise in detecting disease recurrence before clinical relapse, with one study in DLBCL demonstrating a lead time of 3.5 months before imaging confirmed relapse after patients underwent chemoimmunotherapy [30]. 

## 2. Technical Approaches to Molecular Monitoring

### 2.1. Circulating Tumor DNA Quantification Techniques and Technical Considerations

Incorporating liquid biopsy ctDNA-based MRD assessments into clinical practice has the potential to overcome the fundamental limitations of functional imaging. ctDNA refers to the tumor-specific component of cell-free DNA (cfDNA), consisting of DNA fragments released into the bloodstream as cancer cells divide or undergo cell death [31,32]. ctDNA in B-NHLs has been shown to correlate with total MTV, IPI risk score, LDH, Ann Arbor stage, and survival outcomes [33,34,35,36,37,38,39,40,41,42]. Liquid biopsy is able to overcome the challenges of tissue biopsies by non-invasively capturing both spatial and temporal tumor heterogeneity via detection of tumor-specific genetic alterations including single-nucleotide variants (SNVs), chromosomal translocations, insertions/deletions, and copy-number variations (CNVs) [43]. Furthermore, rapid clearance of DNA fragments by DNA nucleases in vivo enables serial ctDNA measurements to inform responsiveness to therapy in real time [35,44]. 

The Continuous Individualized Risk Index (CIRI), a dynamic and personalized outcome prediction model incorporating ctDNA, is under development for DLBCL. CIRI utilizes a ‘win-probability’ model integrating information from pretreatment ctDNA levels, IPI score, and molecular cell of origin (COO) to generate an initial prediction. The model is then updated using risk predictors acquired throughout treatment including molecular response by ctDNA and interim imaging studies. This prediction model has demonstrated superiority to currently utilized individual prognostic variables such as COO, IPI, or interim PET at predicting EFS outcomes at 24 months [45]. Thus, incorporation of ctDNA assessments has the potential to prognosticate, both prior to treatment initiation and throughout the therapeutic continuum, by quantifying depth of remission, monitoring for molecular relapse, and tracking emerging subclones through next-generation sequencing techniques that may be associated with resistance to therapy. 

The most employed ctDNA techniques for MRD detection after CAR-T therapy in B-NHL include polymerase chain reaction (PCR) and next-generation sequencing (NGS). See Table 1 for a comparison of assay methods. Regardless of technique, pre-analytic sample collection and processing must be optimized and standardized to maintain the high sensitivity of assays as the accuracy of downstream molecular analyses is heavily impacted by ctDNA input, which exhibits considerable variation across B-NHL subtypes and stage, and is inherently low at the time of MRD assessment [46]. Thus, a minimum of 10 cubic centimeters (cc) of blood (approximately 4-6 cc of plasma) is recommended to ensure sufficient input of ctDNA [47,48]. Additionally, samples must be collected in either EDTA tubes or in cell-stabilizing tubes and processed within 6 h or 7 days, respectively, to limit contamination of the plasma fraction with peripheral blood mononuclear cell DNA and preserve DNA integrity from nuclease degradation during cell lysis [47,49]. Lastly, the presence of germline variants and accumulation of somatic mutations associated with aging resulting in clonal hematopoiesis of indeterminate potential (CHIP) may lead to background noise confounding accurate detection of tumor-derived somatic mutations [50,51]. However, this can be overcome through parallel sequencing and subsequent subtraction of germline and leukocyte-derived DNA errors [47,52,53,54]. 

### 2.2. Polymerase Chain Reaction-Based Methods

PCR-based assays including allele-specific oligonucleotide quantitative PCR (ASO-qPCR) and digital droplet PCR (ddPCR) can identify and quantify tumor-specific DNA sequences via primers directed to a single or small subset of recurrent somatic variants followed by sequence amplification with sensitivities down to 10^−5^ [55,56]. qPCR employs a fluorescent signal to quantify DNA based on the number of amplification cycles required for detection above background [57]. Conversely, ddPCR utilizes a water–oil emulsion system to partition DNA into thousands of nanoliter-sized droplets where standard PCR amplification occurs, allowing for absolute quantification without the need for a standard curve [58]. Though universal primers exist for disease-defining translocations and mutations, such as t(11;14) in MCL, t(14;18) in FL, and MYD88 mutation in primary central nervous system lymphoma (PCNSL), patient-specific primers developed using the initial tumor sample are required for detection of clonotypic immunoglobulin heavy chain (IgH VDJ) rearrangements. Developing patient-specific primers is a time and labor-intensive process. Furthermore, high rates of somatic hypermutation leading to instability of IgH rearrangements inherent to FL and the germinal center subtype of DLBCL limit the utility of clonotypic-based PCR techniques [59,60]. Thus, PCR-based assays are best suited for MRD detection in B-NHLs with stereotyped translocations or mutations. However, DLBCL is a genetically diverse type of B-NHL, restricting the utility of disease-defining primers in this histologic state [61]. 

### 2.3. Next-Generation Sequencing-Based Methods

NGS-based assays have the potential to overcome the inherent limitations of PCR-based assays and become the new gold standard for ctDNA-based MRD assessments. NGS is a high-throughput, parallel sequencing technology offering deeper and more comprehensive profiling methods to detect genetic alterations with sensitivities as low as 10^−6^ [56]. Broad categories of NGS-based assays used to detect ctDNA include immunoglobulin high-throughput sequencing (IgHTS) and ultrasensitive panel-directed NGS. 

IgHTS-based techniques (i.e., clonoSEQ) utilize universal PCR primers targeting all possible heavy- and light-chain rearrangements followed by NGS of immunoglobulin genes, which permits tracking of both initial clonotypes and emerging subclones. However, this assay still requires a sufficient baseline tumor sample to establish the clonotype. ClonoSEQ is FDA-approved for MRD detection in chronic lymphocytic leukemia (CLL), multiple myeloma (MM), and B-cell acute lymphoblastic leukemia (B-ALL) and has been successfully applied to B-NHL in clinical trials with studies reporting successful clonotype identification and tracking in 100% and 84% of patients with MCL and DLBCL, respectively [30,34,41,42].

Unlike single-gene IgHTS assays, panel-directed NGS techniques (i.e., CAPP-Seq [cancer personalized profiling by deep sequencing]) allow for comprehensive genotyping. Panel-directed NGS assays use libraries and bioinformatic pipelines optimized for low-input DNA to identify frequently occurring tumor-specific SNVs, insertions/deletions, chromosome translocations, and CNVs [43]. Historically high sequencing error rates have been addressed with integrated digital error suppression through the use of molecular barcoding to dramatically improve sensitivity and specificity [62]. Furthermore, panel-directed NGS does not require baseline tumor tissue for patient-specific optimization, as universal sequencing panels target hundreds of lymphoma-specific genetic alterations to capture the breadth of tumor heterogeneity. Excellent concordance has been demonstrated between panel-directed NGS liquid biopsy genotyping and tumor samples in B-NHLs [38,63]. 

Technologies for both IgHTS-based and panel-directed NGS assays for ctDNA assessments in B-NHL are technically complex, requiring calibration and utilization of proprietary bioinformatics pipelines with specialized equipment and expertise [64]. Application of these assays is currently limited to research settings at select academic institutions. 

**Table 1 cancers-16-01881-t001:** Comparison of methods of ctDNA detection for B-NHLs.

	PCR-Based		NGS-Based	
	IgH	Stereotyped Mutations	IgHTS	Panel-Directed
Sensitivity	10^−5^	10^−5^	10^−6^	10^−6^
Requires Primary Tumor Sample	Yes	No	Yes	No
Track Clonal Evolution	No	No	Limited	Yes
Track Resistance	No	No	Limited	Yes
Genotype	No	No	No	Yes
Benefits	Tumor-specific	Tumor-specific Universal primers	Tumor-specific Universal primers Commercially available	Broad genomic coverage Track clonal evolution Track resistance mechanisms
Limitations	Requires primary tumor sample Requires construction of patient-specific primers No genetic information	Knowledge of target mutation required Limited genetic information	Requires primary tumor sample No genetic information	Complex bioinformatics workflow Longer turnaround time Higher cost

## 3. Emerging Applications of ctDNA in the Peri-CAR-T Therapy Setting 

Available literature demonstrates the promise of ctDNA in risk stratification via estimation of tumor burden and genotyping, assessing molecular response to CAR-T therapy, and monitoring for MRD to identify mechanisms of therapy resistance and predict impending clinical relapse. To appreciate the breadth of study heterogeneity, an overview of methods for selected published and emerging studies is summarized in Table 2 and Table 3, respectively. It is worthwhile highlighting differences in sample size, which varies from single digits to over one hundred patients. 

### 3.1. Pre-CAR-T Therapy Risk Stratification

cfDNA assessment at the time of leukapheresis may identify high-risk populations for poor outcomes following CAR-T therapy. Cherng et al. showed that high focal DNA copy-number alteration scores (FCS) obtained using low-pass whole-genome sequencing at leukapheresis in 122 profiled patients with DLBCL predicted inferior 3-month CR rates (28% vs. 56%, *p* = 0.0029) as well as inferior PFS (HR 2.11, *p* = 0.0007) and OS (HR 2.10, *p* = 0.0026). High FCS, indicating genomic instability, was defined as FCS values above the cohort median of 88.5. Deletions in 10q23.31 encoding the FAS death receptor corresponded to poor survival with all patients progressing by 6 months [65]. 

Baseline ctDNA concentration obtained prior to lymphodepleting (LD) therapies is a promising prognostic analyte for CAR-T therapy efficacy outcomes. Several trials within LBCL have demonstrated that lower pre-LD ctDNA concentrations are associated with durable response and improved survival. In the largest study to date (*n* = 138), Sworder et al. demonstrated pre-LD median ctDNA of 21.4 haploid genome equivalents per milliliter (hGE/mL) versus 200 hGE/mL (*p* = 0.005) for durable responders versus progressors, respectively [66]. Similarly, Frank et al. demonstrated that pre-LD ctDNA < 100 LG/mL predicted better PFS and OS compared to ctDNA > 100 LG/mL. ctDNA < 10 LG/mL or between 10 and 100 LG/mL had similar 1-year PFS of 78% and 77% and similar 1-year OS of 90% and 91%, respectively, whereas concentrations between 100 and 1000 and >1000 LG/mL both had median PFS of 3 months, while median OS was 19 months and 7.4 months, respectively [67]. Lastly, Zou et al. reported that low pre-LD ctDNA (*n* = 11) predicted superior 1-year PFS (81.8 vs. 33.3%, *p* = 0.031) and OS (90 vs. 46.7%; *p* = 0.023) [68]. 

Emerging studies in LBCL demonstrated similar findings. Caimi et al. reported patients with pre-LD ctDNA < 250 hGE/mL exhibited higher rates of CR (95% vs. 33%, *p* < 0.01) with longer median PFS (26 vs. 2 months, *p* < 0.001) and OS (NR vs. 20 months, *p* < 0.01) [69]. Foerster et al. showed that high pre-LD ctDNA, defined as mean allele frequency (MAF) ≥ 1%, predicted inferior survival compared to low ctDNA with a median PFS of 86.5 days versus NR (HR 6.3, *p* = 0.04) and OS of 219 days versus NR (HR 4.1, *p* = 0.048) [70]. Lastly, Caldwell et al. reported that baseline ctDNA quantity above the median of the cohort predicted shorter survival with median OS 6.7 months versus NR (*p* = 0.0012) [71]. 

Dean et al. expanded on these findings through incorporation of MTV with thresholds of >147.5 mL and >100 LG/mL to define high MTV and ctDNA, respectively. Patients with low pre-LD ctDNA had better PFS (HR 0.11, *p* < 0.0001) and OS (HR 0.13, *p* = 0.0002). Similarly, patients with low pre-LD MTV had improved PFS (HR 0.17, *p* < 0.0001) and OS (HR 0.19, *p* < 0.0003). Furthermore, this study demonstrated synergistic value of combining ctDNA with MTV to more effectively risk stratify patients. Patients with low MTV/low ctDNA (low-risk group) had better PFS (*p* < 0.0001) and OS (*p* < 0.0001) compared to either low MTV/high ctDNA or high MTV/low ctDNA (intermediate-risk group) or high MTV/high ctDNA (high-risk group) [72]. 

The prognostic value of ctDNA assessments at day 0, defined as the day of CAR-T infusion, has also been reported to correlate with patient outcomes. Sworder et al. found high day 0 ctDNA concentrations were strongly predictive of disease progression and shorter EFS (HR 2.2, *p* = 0.003) and OS (HR 2.1, *p* = 0.014), with median concentrations of 540.4 hGE/mL in progressors versus 11.8 hGE/mL for durable responders [66]. Similarly, Caimi et al. showed that patients with ctDNA < 10 hGE/mL on day 0 had higher CR rates (88% vs. 33%, *p* = 0.03) and longer median PFS (37 vs. 2 months, *p* = 0.002) and OS (NR vs. 13 months, *p* < 0.001) [69]. 

Though additional research is needed to determine the optimal timing of ctDNA assessment prior to CAR-T therapy, these preliminary findings suggest that a pre-LD ctDNA threshold of >100 Lg/mL may effectively identify LBCL patients at high risk for relapse after CAR-T therapy who would benefit from risk-adapted therapeutic approaches to optimize outcomes. Proposed strategies include the addition of bridging therapy or radiation to decrease tumor burden prior to CAR-T therapy, increased ctDNA surveillance after CAR-T therapy to detect subclinical relapse, or reconsideration of CAR-T therapy altogether in patients where risks of cellular therapy outweigh predicted benefits. Prospective trials are required to determine if ctDNA-triggered, risk-adapted therapeutic strategies translate into improved survival outcomes. 

The prognostic value of pre-CAR-T therapy ctDNA assessments within other subtypes of B-NHL is poorly defined due to lack of available data. Further research is needed to validate and harmonize disease-specific ctDNA concentration thresholds dichotomizing high-risk and low-risk groups as shedding of ctDNA varies across B-NHL subtypes. 

### 3.2. Prognostic Role of Tumor Mutational Burden and Evaluation of Emergent Mechanisms of Resistance

Previous studies have illustrated mutational evolution in R/R B-NHL, driven by selective treatment pressures, through the analysis of paired biopsies [73]. In clinical practice, obtaining repeat invasive biopsies at the time of progression or relapse is often impractical or unsafe due to inaccessible tumor sites. Moreover, single-site biopsies may fail to adequately capture spatial tumor heterogeneity, as mutational discordance has been observed between multi-site biopsies [74]. The implementation of panel-based NGS genotyping before CAR-T infusion and after CAR-T therapy failure offers a promising avenue beyond mere MRD detection to deepen our understanding of high-risk genomic signatures and the mechanisms underlying CAR-T therapy resistance.

ctDNA mutational burden has been reported as a potential prognostic variable prior to CAR-T therapy in LBCL with conflicting findings. Zhou et al. demonstrated that fewer ctDNA mutations at the time of initial diagnosis were predictive of durable response and improved OS. The median number of mutations was 3 versus 14.3 in those achieving long-term CR after CAR-T therapy compared to the R/R cohort. Patients carrying <8 mutations at baseline had improved OS compared to those with ≥8 (*p* = 0.014) [75]. Goodman et al. showed that patients achieving CR had significantly lower pre-LD genomic instability number (GIN) levels (*p* = 0.048), a metric capturing copy-number alterations (CNAs) with increased GIN values indicative of increased ctDNA fraction, emergence of tumor-specific CNAs, and increased magnitude of tumor CNAs [76]. Differently, Deng et al. reported that neither the number of mutations nor VAF at day 0, corresponding to the day of CAR-T infusion, was associated with the subsequent response [77]. Differences in reported findings may be due to small sample size as well as the heterogeneity of assessment timing and metrics used to capture mutational burden.

Studies have also harnessed the use of panel-directed NGS to identify genotypes and emerging mutations associated with CAR-T therapy resistance and resultant poor prognosis. Sworder et al. showed that pre-LD mutations in genes shaping B-cell identity (IRF8, PAX5), immunoregulation (TMEM30A), and TP53 were associated with adverse outcomes after CAR-T therapy. Furthermore, emergent genetic alterations were found after subsequent CAR-T therapy failure in various genes including CD19, PPMD1, PAX5, and TP53, as well as amplification of 9p24.1, which codes the immune checkpoint molecules CD274 (PDL-1) and PDCD1LG1 (PD-L2) [66]. Zou et al. identified baseline mutations associated with higher risk of disease progression after CAR-T therapy and resultant inferior PFS and OS including IGLL5, CD79B, P2RY8, ETV6, and KLH6. Interestingly, co-occurrence of TP53 and IGLL5 mutations was observed more frequently in patients with disease progression, suggesting a synergistic role in tumorigenesis and resistance to CAR-T therapy [68]. Lastly, Zhou et al. reported mutations at the time of initial diagnosis in GNA13, SOCS1, TNFAIP3, and XPO1 were associated with R/R disease after CAR-T therapy [75]. 

These findings underscore the role of non-invasive ctDNA-based tumor genotyping to enhance our understanding of genomic determinants of CAR-T therapy resistance, provide insight into tumor evasion mechanisms, guide subsequent treatment strategy, and accelerate the development of novel targeted salvage therapeutics. For example, patients who develop mutations in CD19 or amplification of 9p24.1 would be unlikely to benefit from further T-cell mediated cellular therapies due to loss of CAR-T target and immune evasion, respectively. However, CAR-T combination therapy with immune checkpoint inhibitors could represent a future strategy to overcome immune evasion in 9p24.1 amplified populations. 

### 3.3. Response Assessment and Measurable Residual Disease Detection

cfDNA dynamics after bridging therapy in DLBCL may predict early disease progression after CAR-T therapy. A single study by Bastos-Oreiro et al. demonstrated that ΔcfDNA > 11.5 nanograms/milliliter (ng/mL) between pre-LD and pre-apheresis correlated with 1-month disease progression (*p* < 0.001). This association was maintained after multi-variate analysis (*p* = 0.032). Interestingly, absolute values of cfDNA pre-apheresis and pre-LD were not associated with early progression, suggesting that tumor response to bridging therapy is more informative than tumor burden [78]. 

ctDNA kinetics within the first month after CAR-T infusion are strong predictors of favorable response and survival in LBCL. Study timepoints include 1-week, 2-week, and 1-month post-CAR-T infusion. 

High ctDNA concentrations and poor molecular response at week 1 are associated with poor outcomes after CAR-T therapy in LBCL. Sworder et al. reported that higher median ctDNA concentrations at week 1 (30.4 vs. 0.12 hGE/mL, *p* = 0.003) are associated with disease progression. Furthermore, high ctDNA defined as ≥2.5 log_10_hGE/mL at week 1 predicted inferior EFS (HR 2.0, *p* = 0.004) and OS (HR 1.7, *p* = 0.037) [66]. Similarly, Frank et al. showed that 70% (n = 23) of durably responding patients versus 13% (n = 4) of progressing patients had undetectable ctDNA at 1 week (*p* < 0.0001) [67]. Deng et al. also demonstrated that early molecular response (EMR; >5-fold reduction in VAF relative to D0-1) at day 7 was predictive of ongoing complete response (CR) by PET/CT at 3 months (*p* = 0.008), while no patients with less than a 5-fold molecular response (<5 FMR) achieved CR [77]. Emerging data from Foerster et al. show that patients achieving complete (CMR) and partial metabolic response (PMR) on 4–6-week PET/CT had a significant decrease in ctDNA at days 7–10 compared to those with stable or progressive disease (*p* = 0.001). Furthermore, 100% of patients with >1.5-log-fold reduction in ctDNA on days 7–10 relative to pre-LD achieved CMR or PMR as the best response versus only 62% of patients with less robust ctDNA decline [70]. 

Studies suggest achieving early MRD negativity at week 2 predicts improved outcomes after CAR-T therapy in LBCL. Zou et al. demonstrated that undetectable ctDNA at day 14 predicted improved 3-month CR rates (77.8 vs. 22.2%, *p* = 0.015) [68]. An emerging study by Delfau-Larue et al. showed that 93% (*n* = 13) of patients achieving radiographic CR at 3 months had undetectable ctDNA at day 14 (*p* = 0.009). MRD negativity at day 14 predicted improved 10-month median PFS (NR vs. 7.6 months). Furthermore, day 14 and month 1 ctDNA assessments were concordant in 94% of patients (*n* = 34) [79]. Likewise, preliminary results from the Phase 3 TRANSFORM trial demonstrated that negative day 15 ctDNA was associated with 3-month radiographic CR (*p* = 0.006) [80]. 

Unfavorable ctDNA kinetics at 1 month is associated with poorer outcomes after CAR-T therapy in LBCL. Sworder et al. demonstrated that higher median ctDNA concentrations (7.2 hGE/mL vs. not detected, *p* < 0.001) at week 4 predicted disease progression. ctDNA concentrations ≥ 2.5 log_10_hGE/mL at week 4 were associated with inferior EFS (HR 2.3, *p* < 0.001) and OS (HR 2.5, *p* < 0.001), while patients achieving a major molecular response (MMR; 2.5 log_10_ fold ctDNA decrease relative to D0) at week 4 had statistically significant improvement in EFS (*p* < 0.0001) [66]. Likewise, emerging data from Caldwell et al. show that patients achieving CR/PR (*n* = 21) at day 28 had greater reductions in ctDNA compared to progressors (*n* = 7) with a median mutant molecule per mL (MMPM) log-fold reduction of 1.3 versus −0.9 [71]. 

Persistence of MRD at 1 month predicts poor outcomes after CAR-T therapy in LBCL. Zou et al. demonstrated that achieving ctDNA negativity at day 28 predicted superior 1-year PFS (90.9 vs. 27.3%, *p* = 0.004) and OS (90.9 vs. 49.1%, *p* = 0.003) [68]. Frank et al. showed that the presence of MRD positivity, defined as any detectable ctDNA, at day 28 predicted shorter survival compared to MRD negativity with median PFS of 3.03 months versus NR (*p* < 0.0001) and median OS of 19.0 months versus NR (*p* = 0.0080) [67]. 

Similarly, results from emerging studies in LBCL support the prognostic value of achieving MRD negativity at 1 month. Caimi et al. reported longer median PFS (37 vs. 2 months, *p* = 0.002) and OS (NR vs. 13 months, *p* < 0.001) among patients with undetectable day 30 ctDNA [69]. Similarly, emerging data from the ALYCATE trial showed that 94% (n = 15) of patients achieving radiographic CR at 3 months had undetectable ctDNA at month 1 (*p* = 0.009). MRD negativity at 1 month predicted improved 10-month median PFS (NR vs. 5.8 months) [79]. 

Monitoring ctDNA dynamics has also demonstrated prognostic value after CAR-T therapy in LBCL. Chen et al. demonstrated that peak TP53 mean allele frequency (MAF) > 3.15% within the first trimester after CAR-T therapy is unfavorable and associated with significantly reduced ORR and PFS in patients with TP53 mutated B-NHL. The favorable cohort (defined as peak TP53 MAF < 3.15%) demonstrated an ORR of 92.6% and did not reach median PFS versus ORR of 7.7% and 3 months median PFS for the unfavorable group (*p* < 0.0001). The hazard ratio for disease progression was 19 times higher for the unfavorable cohort (HR 19.45, *p* < 0.0001). First-trimester MAF was found to be a better predictor of PFS than tumor diameter, Ann Arbor stage, IPI risk score, TP53 mutation site, and IgH/MYC translocation [81]. Likewise, Goodman et al. reported that failure to achieve GIN < 170 was associated with eventual disease progression, including in patients achieving CR as the best response. Lower peak GIN values trended toward predicting CR as the best response; however, they did not reach statistical significance (*p* = 0.073), which may be due to insufficient power secondary to small sample size [76]. 

Optimal timepoints for ctDNA assessment after CAR-T therapy within other subtypes of B-NHL are less well defined due to the paucity of available research. In MCL, the ZUMA-2 trial demonstrated strong predictive performance of MRD monitoring at 3 months and 6 months in estimating risk of relapse, with area under the curve (AUC) of 0.8000 and 0.7500, respectively. In the MRD-evaluable cohort, MRD positivity at 6 months was associated with inferior DOR (6.1 months vs. NR), PFS (7.1 months vs. NR), and OS (27.0 months vs. NR) compared to MRD negativity [82]. Ananth et al. reported no significant differences in PFS or OS in MCL patients with detectable MRD at day 28 [83]. Jimenez-Ubieto et al. suggested that optimal timepoints for MRD assessment in FL may be between week 1 and 3 months. Despite all patients achieving CR as the best response, MRD positivity was maintained at day 7 for all patients while all durably responding patients had undetectable ctDNA nearly three months after CAR-T infusion [84]. 

Post-treatment kinetics and MRD assessments represent emerging clinical trial endpoints and potential surrogates for survival. Patients with unfavorable molecular response or MRD positivity signal populations at higher risk for relapse that may benefit from enhanced ctDNA surveillance to detect early progression. Further, those with persistent MRD may benefit from the addition of consolidative therapies to deepen response or delay maintenance therapy to control molecular disease burden and prolong the time to clinical relapse and the need for subsequent lines of intensive therapy. 

### 3.4. ctDNA Concordance with Post-CAR-T Therapy PET/CT

Overall, studies support that ctDNA is concordant with the response on PET/CT in LBCL. Frank et al. showed that clonoSEQ MRD assessments for patients with partial response (PR) or stable disease (SD) on PET/CT at day 28 had a sensitivity of 94%, specificity of 82%, positive predictive value (PPV) of 88%, and negative predictive value (NPV) of 90% [67]. Similarly, Zhou et al. demonstrated comparable sensitivity (94.7%) and specificity (83.3%) of panel-based NGS ctDNA profiling in detecting MRD [75]. Hossain et al. and Caldwell et al. both reported a strong correlation between changes in ctDNA concentration and MTV from pre-CART to day 28, suggesting ctDNA concentrations are reflective of metabolically active tumor burden [71,85]. 

An emerging study from Miles et al. suggests that the number of lines (#L) of therapy prior to CAR-T therapy may impact ctDNA concordance with PET/CT in patients with DLBCL. The PPV of day 28 clonoSEQ ctDNA assessment in 1L versus 3L cohorts was 50% (*n* = 1) versus 88% (*n* = 7), respectively; whereas NPV was 88% (*n* = 7) versus 83% (*n* = 10), respectively [86]. 

Valuable insight into treatment response and efficacy outcomes can still be gained at times of discordance between ctDNA and functional imaging assessments. For example, Dean et al. showed that after excluding MRD-negative patients achieving CR, ctDNA and MTV correlated at 3 months (rs, 0.79; *p* = 0.0007), but not at 1 month (rs, 0.28; *p* = 0.11). The presence of FDG-avid lesions on 1-month PET/CT with undetectable ctDNA suggests an ongoing treatment response rather than refractory disease or progression necessitating a change of therapeutic strategy. Additionally, though 71% of patients in radiographic CR had undetectable ctDNA at paired assessment, the TRANSFORM trial demonstrated detectable ctDNA predicts shorter EFS when ctDNA and PET/CT responses are discordant at 1-month (*p* = 0.011), 3-month (*p* = 0.025), and 12-month (*p* = 0.003) assessments [80].

These findings support the use of ctDNA as a complementary analyte capable of indirectly measuring the metabolically active tumor. Furthermore, concurrent ctDNA assessment informs the interpretation of PET/CT imaging. ctDNA adds specificity in delineating between pseudoprogression versus CAR-T therapy failure on 1-month PET/CT and improves the sensitivity of detecting residual disease at times of radiographic CR, with positive ctDNA signaling patients at higher risk of poor survival outcomes. 

### 3.5. Detection of Molecular Relapse

ctDNA monitoring after CAR-T therapy increases lead time detection of disease relapse compared to conventional PET/CT imaging across B-NHL subtypes. Chen et al. reported that mean lead time for detection of disease progression with ddPCR in DLBCL was 1.3 months (range 1–2) [81], while Ananth et al. reported clonoSEQ predicted relapse ahead of PET/CT in 86% (*n* = 6) of patients with a median lead time of 294 days (range 15–332) in patients with MCL [83]. Miles et al. reported lead time detection may vary across the number of lines of therapy in patients with DLBCL assessed using clonoSEQ. All patients (*n* = 5) status post three lines of therapy prior to CAR-T therapy had detectable ctDNA by the time of radiographic progression with a median lead time of 43 days. Differently, only 78% (*n* = 7) of patients’ status post two lines of therapy had detectable ctDNA by the time of radiographic progression with median lead time of 35 days [86]. The dramatic variation in lead time detection observed between DLBCL versus MCL studies may be explained by variation in pre-analytic factors, including ctDNA shedding, and improved sensitivity of NGS-based assays compared to PCR-based.

While other studies in this review did not quantify lead times, molecular relapse was detected prior to radiographic relapse with few exceptions. For example, Frank et al. and Jimenez-Ubieto et al. reported that ctDNA was detected at or before radiographic relapse in 94% (*n* = 29) with LBCL [67] and 100% (*n* = 2) [84] of progressing patients with MCL, respectively, while Goodman et al. reported increasing GIN was observed before radiographic progression in 83% (*n* = 5) of patients with LBCL [76]. Likewise, Hossain et al. reported rising ctDNA preceded radiographic disease progression in 80% (*n* = 4) of patients with DLBCL, while 100% had rising ctDNA at the time of radiographic relapse (*n* = 5) [85].

Serial ctDNA assessments represent a novel approach to surveillance after CAR-T therapy offering an opportunity for early therapeutic interventions ahead of clinical relapse. However, it remains unknown if early interventions at the time of molecular relapse translate into improved clinical outcomes. Furthermore, standardized assay methods and definitions for molecular relapse are needed. In populations at elevated risk for relapse after CAR-T therapy, ctDNA allows for shortened examination intervals; however, optimal assessment frequency is currently unknown. Once ctDNA concentrations cross a yet-to-be-determined positivity threshold, reflexive PET/CT imaging could be obtained to assess for radiographic evidence of relapse. To realize the potential of ctDNA-based surveillance in clinical practice, early detection of subclinical disease and subsequent treatment strategies aimed at eradicating molecular relapse must lead to improved outcomes. 

**Table 2 cancers-16-01881-t002:** Published studies on ctDNA for prognosis, response assessment, and surveillance in B-NHL after CAR-T therapy.

Study	R/R Disease	CAR-T Product	N	Disease Assessments	Timing	Plasma Volume	Sensitivity
Circulating tumor DNA assessments in patients with diffuse large B-cell lymphoma following CAR-T therapy [85] Hossain et al. *Leukemia & Lymphoma* 2019	DLBCL	Axicabtagene ciloleucel	6	1. IgHTS 2. PET/CT (MTV)	1. Pre-CART infusion and days 0, 7, 14, 21, 28, 56, 90 2. Pre-CART infusion, day 28, and month 3	N/A	10^−6^
Characteristics of anti-CD19 CAR T cell infusion products associated with efficacy and toxicity in patients with large B cell lymphomas [77] Deng et al. *Nature* 2020	DLBCL, tFL, PMBCL, HGBCL	Axicabtagene ciloleucel	24	1. Panel-directed NGS 2. PET/CT	1. Days 0–1, weeks 1, 2, and 4 2. Month 3	2 mL	N/A
TP53-mutated circulating tumor DNA for disease monitoring in lymphoma patients after CAR T cell therapy [81] Chen et al. *Diagnostics* 2021	TP53 mutated B-NHL	CD19 and CD20 CAR-T NOS	40	1. ddPCR	Median follow up 3 months Mean follow up 6.35 months	Median 4.8 mL (3.5–7.0 mL)	10^−4^
Monitoring of circulating tumor DNA Improves early relapse detection after axicabtagene ciloleucel infusion in large B-cell lymphoma: results of a prospective multi-institutional trial [67] Frank et al. *Journal of Clinical Oncology* 2021	DLBCL	Axicabtagene ciloleucel	57	1. IgHTS 2. PET/CT	1. Pre-LD and days 0, 7, 14, 21, 28, 56, 90, 180, 365 2. Pre-apheresis and months 1, 3, 6, and 12	8.5 mL	10^−5^
Determinants of resistance to engineered T cell therapies targeting CD19 in lymphoma [66] Sworder et al. *Cancer Cell* 2021	LBCL	Axicabtagene ciloleucel	138	1. Panel-directed NGS	1. Pre-LD and days 0, 7, 28	2–6 mL	10^−6^
Serial surveillance by circulating tumor DNA profiling after chimeric antigen receptor T therapy for the guidance of r/r diffuse large B cell lymphoma precise treatment [75] Zhou et al. *Journal of Cancer* 2021	DLBCL	CD19 CAR-T NOS	8	1. Panel-directed NGS 2. PET/CT	1. Retrospective: initial diagnosis, at relapse, and at time of progression after CAR-T 2. Retrospective	8–10 mL of PB	N/A
Assessing CAR T-cell therapy response using genome-wide sequencing of cell-free DNA in patients with B-cell lymphomas [76] Goodman et al. *Transplant and Cellular Therapy* 2022	HGBCL DLBCL THRLBCL	Axicabtagene ciloleucel, Tisagenlecleucel	12	1. Genome-wide NGS 2. PET/CT	1. Days −5, 0, 1, 3, 5, 7, 14, 21, 28 and every 4 to 8 weeks 2. Discretion of treating physician	N/A	N/A
Risk assessment with low-pass whole-genome sequencing of cell-free DNA before CD19 CAR T-cell therapy for large B-cell lymphoma [65] Cherng et al. *Blood* 2022	DLBCL, transformed indolent, PMBCL	Axicabtagene ciloleucel, Tisagenlecleucel	122	1. Low-pass WGS 2. PET/CT	1. Leukapheresis 2. Months 1 and 3	4–10 mL	N/A
Circulating tumor DNA adds specificity to PET after axicabtagene ciloleucel in large B-cell lymphoma [72] Dean et al. *Blood* 2023	LBCL	Axicabtagene ciloleucel	72	1. IgHTS 2. PET/CT (MTV)	1. Pre-LD and days 7, 14, 21, 56, 180, 270, 300, 360 2. Pre-LD and months 1 and 3	N/A	10^−6^
Cell-free DNA Dynamic concentration and other variables are predictors of early progression after chimeric antigen receptor T cell therapy in patients with diffuse large B cell lymphoma [78] Bastos-Oreiro et al. *Transplant and Cellular Therapy* 2023	DLBCL, tFL, PMBCL	Axicabtagene ciloleucel, Tisagenlecleucel	58	1. Panel-directed NGS 2. PET/CT	1. Retrospective: Pre-apheresis and pre-LD 2. Pre-apheresis, pre-LD, and months 1 and 3	5–10 mL	N/A
Dynamic monitoring of circulating tumor DNA reveals outcomes and genomic alterations in patients with relapsed or refractory large B-cell lymphoma undergoing CAR T-cell therapy [68] Zou et al. *Journal of Immunotherapy Cancer* 2024	DLBCL, HGBCL, tFL	Axicabtagene ciloleucel, CNCT19	23	1. Panel-directed NGS 2. PET and/or CT	1. Pre-LD and days 14, 28, 60, 90 and beyond 120 or at progression 2. Days 30, 90, and every three months thereafter	N/A	N/A
Personalized monitoring of circulating tumor DNA with specific signature of trackable mutations after chimeric antigen receptor T-cell therapy in follicular lymphoma patients [84] Jimenez-Ubieta et al. *Frontiers in Immunology* 2023	FL	CD19 CAR-T NOS	10	1. Panel-directed NGS 2. PET/CT	1. Pre-LD, days 7 and 28, and months 3, 6, 12, 24, 36 2. Pre-LD and days 90, 180, 365 and every 6 months	10–20 mL of PB	10^−4^

R/R relapse/refractory, N number, DLBCL diffuse large B-cell lymphoma, tFL transformed follicular lymphoma, PMBCL primary mediastinal B-cell lymphoma, HGBCL high-grade B-cell lymphoma, LBCL large B-cell lymphoma, THRLBCL T-cell/histiocyte-rich large B-cell lymphoma, FL follicular lymphoma, NGS next-generation sequencing, WGS whole-genome sequencing, NOS not otherwise specified, MTV metabolic tumor volume, pre-LD pre-lymphodepletion, N/A not available, PB peripheral blood.

**Table 3 cancers-16-01881-t003:** Emerging abstracts on ctDNA for prognosis, response assessment, and surveillance in B-NHL after CAR-T therapy.

Study	R/R Disease	CAR-T Product	N	Disease Assessments	Timing	Plasma Volume	Sensitivity
Circulating tumor DNA correlation with lymphoma response and survival outcomes at multiple time points of anti-CD19 CAR T-cell therapy [69] Caimi et al. ASH Abstract 2022	B-NHL	CD19 CAR-T NOS	28	1. Panel-directed NGS 2. PET/CT	1. Pre-LD and days 0, 30 2. Days 30 and 90	N/A	10^−6^
Early prediction of treatment response by circulating tumor DNA profiling in patients with diffuse large B-cell lymphoma receiving CAR T-cell therapy [70] Foerster et al. ASH Abstract 2023	DLBCL	Axicabtagene ciloleucel, Tisagenlecleucel	16	1. Panel-directed NGS 2. PET/CT	1. Pre-LD, and days 7–10 2. Weeks 4–6	N/A	N/A
Early ctDNA clearance after CAR T-cell infusion predicts outcomes in patients with large B-cell lymphoma: results from ALYCANTE, a phase 2 lysa study [79] Delfau-Larue et al. EHA Abstract 2023	LBCL	Axicabtagene ciloleucel	62	1. Panel-directed NGS	1. Leukapheresis, pre-LD, days 0 and 14, months 1, 3, 6, 9, and 12	2–4 mL	N/A
Circulating tumor DNA (ctDNA) by clonoSEQ to monitoring residual disease after axicabtagene ciloleucel (axi-cel) in large B-cell lymphoma (LBCL) [86] Miles et al. ASCO Abstract 2023	LBCL	Axicabtagene ciloleucel	N/A	1. IgHTS	1. 1L: Day 28, months 3 and 6 2L: Days 50, 100, 150, months 9 and 24 3L: Day 28, months 3 and 5	N/A	10^−5^
Circulating tumor DNA dynamics as early outcome predictors for Lisocabtagene Maraleucel as second line therapy for large B-cell lymphoma from the Phase 3 TRANSFORM [75] Stepan et al. ASH Abstract 2023	LBCL	Lisocabtagene maraleucel	7	1. Panel-directed NGS 2. PET/CT	1. Baseline (study screening), day 0 and 15, and months 1, 2, 3, and 12 2. N/A	N/A	10^−6^
Clinical utility of circulating tumor (ct)DNA quantity and kinetics in patients undergoing therapy for relapsed/refractory aggressive B-cell lymphoma [71] Caldwell et al. ASH Abstract 2023	DLBCL HGBL PMBL tFL MZL	Axicabtagene ciloleucel, Tisagenlecleucel	28	1. Panel-directed NGS	1. N/A	N/A	N/A
Post-CAR-T minimal residual disease monitoring in mantle cell lymphoma enables early relapse detection [83] Ananth et al. ASH Abstract 2023	MCL	Brexucabtagene autoleucel	34	1. IgHTS	1. Months 1, 3, 6, and 12	N/A	10^−6^

R/R relapse/refractory, N number, DLBCL diffuse large B-cell lymphoma, LBCL large B-cell lymphoma, tFL transformed follicular lymphoma, PMBCL primary mediastinal B-cell lymphoma, HGBCL high-grade B-cell lymphoma, MZL marginal zone lymphoma, MCL mantle cell lymphoma, NOS not otherwise specified, N/A not available, NGS next-generation sequencing, IgHTS immunoglobulin high-throughput sequencing, pre-LD pre-lymphodepletion, 1L status post one line of treatment, 2L status post two lines of treatment, 3L status post three lines of treatment.

## 4. Future Directions

### 4.1. STEPing beyond ctDNA

The simultaneous profiling of tumor-intrinsic and -extrinsic effector T-cell molecular features using NGS technology represents the future of precision medicine in the era of cellular therapeutics to predict CAR-T therapy outcomes and personalize subsequent treatment strategies.

Sworder et al. showed that elevated levels of cfCAR19 at week 1, a timepoint reflecting peak antigen-driven CAR-T expansion, correlated to improved EFS (HR 0.55, *p* = 0.010). The Stanford group subsequently developed the Simultaneous Tumor and Effector Profiling (STEP) platform, a novel multivariable prediction tool integrating complementary analysis of ctDNA and cell-free CAR19 DNA (cfCAR19) at week 4 and week 1, respectively. Discovery and validation cohorts demonstrated that high STEP scores predict inferior EFS (HR 2.7, *p* < 0.001) and OS (HR 5.4, *p* < 0.001) in patients with DLBCL treated with axi-cel [66].

Deng et al. demonstrated that infusion products enriched with CD8 memory T-cell phenotype correlated with ongoing CR at 3 months (q = 1.79 × 10^−33^), whereas patients with PR/PD at 3 months exhibited increased CD8 T-cell exhaustion signatures (q = 1.96 × 10^−45^). Patients failing to achieve greater than a 5-fold molecular response (>5 FMR) by day 7 demonstrated higher fraction of LAG3 and TIM3 co-expression by CD8 T-cells (q < 2.38 × 10^−170^), which function as co-inhibitory molecules limiting T-cell activation and expansion [77]. 

These findings suggest that optimization of CAR-T expansion through increasing the dose of infused cells, number of infusions, enrichment of CD8 memory T-cell phenotype during manufacturing, or targeted therapy directed toward markers associated with CAR-T exhaustion such as LAG3 or TIM3 may be beneficial. Additionally, the persistence of cfCAR19 at relapse suggests the patient may not benefit from further T-cell-mediated treatments and alternate therapies should be considered.

### 4.2. Barriers in Bringing ctDNA from Bench to Bedside

ctDNA has emerged as a promising complementary prognostic biomarker with various clinical applications including pre-CAR-T therapy risk stratification, assessment of treatment response, early detection of molecular relapse, and comprehensive genotyping to aid in the identification of therapy resistance. However, harnessing the potential for ctDNA to advance personalized medicine in the era of cellular therapeutics first necessitates validation of ctDNA as a replicable analyte across clinical trials. The current literature exploring ctDNA applications in the peri-CAR-T therapy setting remains in its infancy, comprised primarily of small proof-of-concept and hypothesis-generating studies within LBCL. Furthermore, substantial heterogeneity exists between trial methods regarding liquid biopsy sample type and volume, subsequent molecular assays and bioinformatic workflows, timing of ctDNA assessments, and reported response metrics. Standardization of de-centralized pre-analytic processing and harmonization of molecular detection techniques and clinical response metrics is needed to allow for inter-study comparisons. Only then can consensus guidelines be established to define appropriate clinical scenarios for ctDNA evaluations, optimal assessment intervals, and definitions of molecular response/relapse [47]. 

Knowledge gaps regarding how ctDNA should be incorporated into clinical decision making remain. While ctDNA is likely to be helpful in differentiating between pseudoprogression versus true disease progression without the need for confirmatory biopsy, it is unknown whether ctDNA-guided treatment strategies following CAR-T therapy, such as incorporation of consolidative therapy in those deemed high risk for early relapse or initiation of maintenance therapy at the time of molecular relapse, are superior to the current standard of care or merely add toxicity without improving survival. Prospective multi-center ctDNA-adapted clinical trials including a wider breadth of B-NHL subtypes are needed to realize the potential of ctDNA in clinical practice. It is worthwhile to highlight that even once ctDNA is approved for clinical use in B-NHL, affordability and accessibility of testing will remain major concerns that may limit widespread adoption [87]. 

## 5. Conclusions

The integration of ctDNA assessment into the management of patients with B-NHL undergoing CAR-T therapy represents the dawn of an advancement in precision medicine. ctDNA has untapped potential to alter paradigms in risk stratification, therapeutic strategies, and surveillance in relapsed/refractory B-NHL. By providing real-time insights into treatment response and emergent mechanisms of resistance to therapy, ctDNA portends a valuable addition to our diagnostic arsenal. However, validation of ctDNA as a prognostic analyte and surrogate endpoint requires collaborative efforts to generate sufficiently powered prospective clinical trials before ctDNA is ready for routine use in clinical practice. With further research and refinement, ctDNA stands poised to advance the field of cellular therapeutics in lymphoma with the goal of guiding risk- and response-adapted therapeutic strategies to improve patient outcomes.

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
