# Peer review of "Circulating Tumor DNA as a Complementary Prognostic Biomarker during CAR-T Therapy in B-Cell Non-Hodgkin Lymphomas"

_cancers, 2024, doi:10.3390/cancers16101881_

Round 1

Reviewer 1 Report

Comments and Suggestions for Authors

Monick and Rosenthal provide a well-structured and written overview on the use of cfDNA as a biomarker for prognostication in patients with B-cell lymphoma treated with CAR-T cells. The authors performed a very comprehensive overview of the subject, with a critical appraisal of published results. The tables are very informative and useful for the reader.

Major issues:

There are some publications addressing the theme of the review which have not been cited but that should be considered:

Zou et al, 2024, doi: 10.1136/jitc-2023-008450

Wittibschlager et al, doi: 10.3390/ijms24065688.

Bastos-Oreiro et al, doi: 10.1016/j.jtct.2023.03.009.

Cherng et al, doi: 10.1182/blood.2022015601.

Minor issues:

Line 152: Please spell-out “PCNSL”

Lines 208-209, 297-299, 409-420: please format text adequately.

Author Response

  1. There are some publications addressing the theme of the review which have not been cited but that should be considered: Zou et al, 2024, doi: 10.1136/jitc-2023-008450, Wittibschlager et al, doi: 10.3390/ijms24065688., Bastos-Oreiro et al, doi: 10.1016/j.jtct.2023.03.009., Cherng et al, doi: 10.1182/blood.2022015601.

Thank you for bringing these publications to our attention. We have included discussions of Zou et al., Bastos-Oreiro et al., and Cherng et al. We did not include Wittibschlager et al., in the review as this paper focuses on CAR-T persistent at 6 months (measured using cfDNA) and patient outcomes. This paper did not incorporate cfDNA or ctDNA of B-NHL with CAR-T persistent or outcomes. We thus we feel that it is out of the scope for this review paper.

  1. Line 152: Please spell-out “PCNSL”

Primary central nervous system lymphoma has been spelled out.

  1. Lines 208-209, 297-299, 409-420: please format text adequately.

Text has been correctly formatted (uniform font and size)

Reviewer 2 Report

Comments and Suggestions for Authors

Monique and Rosenthal's manuscript summarizes recent advances in the use of liquid biopsy as an additional prognostic biomarker during CAR-T cell therapy for B-cell non-Hodgkin's lymphomas.  Assessing disease-associated DNA changes using ctDNA can provide valuable information on response dynamics, molecular mechanisms of resistance, and early detection of molecular relapse, which has great clinical potential by providing a minimally invasive tool for clinical practice. By reviewing recent literature, the authors show that the translational ability of assessing molecular biomarkers in ctDNA after CAR-T cell therapy may represent the future of precision medicine for predicting CAR-T outcomes and personalizing subsequent treatment strategies.

Authors conclude that the integration of ctDNA-based biomarker assessment into the management of B-cell non-Hodgkin lymphomas (B-NHL) undergoing CAR-T therapy represents a transformative advancement in precision medicine. By providing real-time insights into minimal residual disease (MRD) monitoring and treatment response, ctDNA represents a valuable addition to our diagnostic arsenal.

 Comparing the results obtained in various studies evaluating ctDNA-based biomarkers, the authors conclude that validation of ctDNA as a predictive analyte and surrogate endpoint requires a collaborative effort in prospective clinical trials.

The review provides a comprehensive analysis of studies demonstrating the ability of ctDNA to be a disease-specific biomarker by applying an appropriate molecular tool to analyze changes in DNA levels.

Minor remarks

Some abbreviation used were not define explicitly:

Line 132 cc

Line 197 ctDNA >100 LG/mL and so on.

In many sentences after CAR-T… should be written cell therapy.

The font of lines 409-420 has been changed.

In general, some paragraphs are overloaded with statistical data, which in some way makes it difficult to perceive an article

The manuscript is well written in English, but appears to require some work.

Comments on the Quality of English Language

The manuscript is well written in English.

Author Response

  1. Some abbreviations used were not defined explicitly: Line 132 cc, Line 197 ctDNA >100 LG/mL and so on

Units of measurements have been explicitly defined.

  1. In many sentences after CAR-T… should be written cell therapy.

Therapy has been included after CAR-T. We defined CAR-T to represent “chimeric antigen receptor T-cell," thus we feel that including cell before therapy is redundant.

  1. The font of lines 409-420 has been changed.

Text has been correctly formatted (uniform font and size)

  1. In general, some paragraphs are overloaded with statistical data, which in some way makes it difficult to perceive an article.

There is significant heterogeneity in the cited studies, both in methodologies and sample sizes. As such, we felt the need to highlight and correlate the statistical aspects of these studies with their clinical significance.

Reviewer 3 Report

Comments and Suggestions for Authors

This is a very interesting report of all published studies examining the prognostic value of circulating tumor DNA in pre- and post-CAR-T settings, with a careful overview and comparison of the limitations and advantages of different detection methods and time points. The identification of a reliable biomarker is still a long way off, this is the final message of your investigation. Your manuscript shows some discordant results and the need for standardization of techniques and time points adapted to clinical characteristics, such as the number of previous therapies, IPI-score as pre-CAR-T variables, and PET results after CAR-T. I have only a few observations:

Page 4, line 152: t(14;18) and not t(11;14) is the FL-defining translocation.

Table 2 shows a wide range in the number of patients analyzed in the different studies. I believe that studies with a low number of patients may have less statistical power than larger studies. I think the authors should highlight this when describing the results in the text or, if possible, place more emphasis on larger studies.

Page 12, line 449-451: please provide the definition of high STEP score.

Author Response

  1. Page 4, line 152: t(14;18) and not t(11;14) is the FL-defining translocation.

t(11;14) has been corrected to correctly represent FL defining translocation

  1. Table 2 shows a wide range in the number of patients analyzed in the different studies. I believe that studies with a low number of patients may have less statistical power than larger studies. I think the authors should highlight this when describing the results in the text or, if possible, place more emphasis on larger studies.

We have highlighted Table 2 and 3 in an introductory paragraph to draw readers attention to the heterogeneity of study design, including sample size. We have also highlighted larger studies within the text.      

  1. Page 12, line 449-451: please provide the definition of high STEP score.

Upon re-review of Sworder et al. manuscript and supplementary information, no definition for “high” STEP score was provided. We are unfortunately unable to include this information.